# STING is redundant for host defense and pathology of COVID-19-like disease in mice

Giorgia Marino[1], Baocun Zhang[1], Alexander Schmitz[1], Hanna VF Schwensen[2], Line S Reinert[1],*, Søren R Paludan[1],*

**Critical COVID-19 is characterized by lack of early type I interferon-mediated host defense and subsequent hyper-inflammation in the lungs. Aberrant activation of macrophages and neutrophils has been reported to lead to excessive activation of innate immunological pathways. It has recently been suggested that the DNA-sensing cGAS–STING pathway drives pathology in the SARS-CoV-2–infected lungs, but mechanistic understanding from in vivo models is needed. Here, we tested whether STING is involved in COVID-19-like disease using the K18-hACE2 mouse model. We report that disease development after SARS-CoV-2 infection is unaltered in STING-deficient K18-hACE2 mice. In agreement with this, STING deficiency did not affect control of viral replication or production of interferons and inflammatory cytokines. This was accompanied by comparable profiles of infiltrating immune cells into the lungs of infected mice. These data do not support a role for STING in COVID-19 pathology and calls for further investigation into the pathogenesis of critical COVID-19.**

## Introduction

Coronavirus disease (COVID-19) is caused by SARS-CoV-2, which emerged in late 2019 to cause a pandemic in 2020, with millions of cases of critical disease and death. SARS-CoV-2 infects primarily cells in the respiratory tract to cause a broad range of outcomes from abortive infections, via mild cold-like symptoms, to severe respiratory disease and death (Sherwani & Khan, 2020). For efficient SARS-CoV-2 infection of the airways, there is requirement for both the receptor angiotensin-converting enzyme 2 (ACE2) and the transmembrane protease, serine 2 (TMPRSS2). SARS-CoV-2 enters into cells using the spike protein (S), which is cleaved into the S1 and S2 subunits, by TMPRSS2 or furin-like enzymes, depending on the sequence of spike at the S1/S2 junction. The latter engages ACE2 as a receptor for S1 (Mehta et al, 2020; Iwata-Yoshikawa et al, 2022; Jackson et al, 2022).

Innate immune responses, most notably the type I interferon (IFN-I) system, play a central role in early control of virus infections, including SARS-CoV-2 (Bastard et al, 2020; Zhang et al, 2020). However, this part of the immune system can also contribute to disease development. In fact, the hyper-inflammatory response is a central player in critical COVID-19 (Paludan & Mogensen, 2022), which can lead to a multi-organ failure and death (Yang et al, 2020). As the disease progresses, there is recruitment of leukocytes, including neutrophils and macrophages, which in turn express pro-inflammatory mediators, including the cytokines interleukin (IL)-1$\beta$, IL-6, TNF$\alpha$, and IFN-I, and the chemokine CXCL10 (Han et al, 2020). In the case of critical COVID-19, accelerating activation of some inflammatory processes contributes to exhaustion of specific lymphocyte subsets and promotes cell death pathways (Paludan & Mogensen, 2022). Therefore, there is an interest in identifying innate immunological pathways that contribute to host defense and COVID-19 pathogenesis. The DNA-activated cGAS–STING pathway was recently suggested to both drive an inflammatory response in SARS-CoV-2–infected cells and to contribute to disease pathogenesis (Domizio et al, 2022; Neufeldt et al, 2022). These studies suggested contributions from both the transcription factor nuclear factor $\kappa$B and the IFN-I system (Domizio et al, 2022; Neufeldt et al, 2022), and that the DNA-triggering activation of the pathway was derived from damaged mitochondria releasing DNA in the cytosol, triggering cGAS–STING signaling (Domizio et al, 2022). In contrast to these data, other studies have used synthetic agonists for STING to demonstrate a protective role for this pathway, mainly in the context of immunotherapy (Humphries et al, 2021; Li et al, 2021). These results suggest a complex role for the cGAS–STING pathway in COVID-19 and urges for mechanistic studies in model systems.

## Results and Discussion

Critical COVID-19 is characterized by excessive activation of innate immune responses (Paludan & Mogensen, 2022). The K18-hACE2 transgenic mouse model shows key features of critical COVID-19, including viral replication in the lungs, strong inflammatory

---

[1]Department of Biomedicine, Aarhus University, Aarhus, Denmark   [2]Department of Histopathology, Aarhus University Hospital, Aarhus, Denmark

Correspondence: srp@biomed.au.dk
*Line S Reinert and Søren R Paludan contributed equally to this work and shared last author

disease, damage of lung tissue and integrity, thrombosis, vasculitis, and age-dependent susceptibility to lethality (Bao et al, 2020; Yinda et al, 2021; Zheng et al, 2021; Lee et al, 2022; Tiwari et al, 2022). In addition, the model also recapitulates some post-infected features of COVID-19, including anosmia (Zheng et al, 2021). The limitation of this model includes higher hACE2 expression in the brain of K18-hACE2 mice compared with the human brain (Dong et al, 2022), thus leading to neuro-invasion to an extent exceeding what has been observed in humans (Kumari et al, 2021). Despite these differences, this mouse model mimics many of the clinical features of SARS-CoV-2 infection and COVID-like disease (Arce & Costoya, 2021).

To study the role of STING in a COVID-19-like disease, we first challenged K18-hACE2 mice intranasally with SARS-CoV2 and tissues were harvested at different timepoints after infection. In agreement with findings in humans (Jalloh et al, 2022; Neufeldt et al, 2022), and supporting the hypothesis of a role for STING in COVID-19 pathology, we observed the pathway to be activated in SARS-CoV-2-infected lungs, as measured by accumulation of STING phosphorylated at residue serine 365 (corresponding to human serine 366) (Fig 1A). To mechanistically explore the role of the cGAS–STING pathway in disease pathology, we crossed STING-deficient Gold-enticket mice with K18-hACE2 transgenic mice (Fig S1A). Interestingly, the STING-deficient K18-hACE2 mice showed no difference in weight change and survival following infection were comparable with control K18-hACE2 mice (Fig 1B and C). No differences between male and female mice were observed (Fig S1B–E). Similar results were obtained when using mice heterozygous for *Sting* (Fig S1F and G). TCDI50 assay and qRT-PCR analysis for viral load and transcripts showed that viral replication was not affected by STING deficiency (Fig 1D and E). Consistent with reports from others (Domizio et al, 2022), we found elevated levels of *Retnla* and F3 upon SARS-CoV-2 infection (Fig 1F and G), which are markers of lung damage and loss of tissue integrity. However, no significant difference between the STING-proficient and -deficient mice was observed. These findings were also confirmed by pathology examination of the lungs. The elevation of neutrophil infiltration, hemorrhage, and thrombosis in the infected lungs were not affected by STING deficiency (Figs 1H–J and S1H).

The first set of experiments was performed using the SARS-CoV-2 alpha variant. To test whether the same was observed with other viral variants, we infected K18-hACE2 Sting$^{gt/gt}$ mice with delta and omicron variants (Fig 2A and B). As reported by others, the omicron variant was less pathogenic than earlier variants (Bentley et al, 2021 *Preprint*; Natekar et al, 2022), but no effect of STING deficiency was observed (Fig 2A and B). The SARS-CoV-2 delta variant can infect mice and cause some degree of disease (Lee et al, 2022). In this model, we also observed no effect of STING deficiency (Fig 2C–E).

To investigate whether STING deficiency impacted cytokine expressions in the lungs, qRT-PCR analysis of lung tissues was performed. In agreement with previous reports (Shibabaw et al, 2020; Winkler et al, 2020), SARS-CoV-2 infection induced expression of *Ifnb*, inflammatory cytokines, and chemokines in the lungs (Fig 3). However, among most of the transcripts examined, we observed no significant difference in these gene expressions between K18-hACE2 and K18-hACE2 Sting$^{gt/gt}$ mice. The only exception being TNF Receptor Superfamily Member 12A (Tnfrsf12a) mRNA expression levels, which were constitutively lower in STING-deficient K18

h-ACE2 mice (Fig 3J). Protein analysis of CXCL10 form the lungs homogenate verified that the lack of difference in the expression of cytokines and chemokines between K18-hACE and STING-deficient K18 h-ACE2 mice was also seen at the protein level (Fig 3O).

To further examine the role of STING in COVID-19-like disease in K18-hACE2 mice, we used suspension mass cytometry analysis to identify and distinguish a panel of immune cell populations (Fig S2). When analyzing cells isolated from the lungs on day 6 p.i., we found that STING deficiency led to only minor changes in the leukocyte's composition in the infected lungs (Fig 4), with notably most macrophage populations and neutrophil levels being unaltered (Fig 4K). We did observe a decrease in nonclassical monocytes and macrophages Mo/Mφ (Ly6C⁻ CD43⁺) and interstitial macrophages (IMφ) in the SARS-CoV-2-infected STING-deficient lungs (Fig 4C and D). These two populations have been proposed to be ontologically linked (Liegeois et al, 2018). Although high abundance of inflammatory macrophages is a key feature of severe COVID-19 (Liao et al, 2020; Melms et al, 2021; Ren et al, 2021; Wauters et al, 2021), depletion of macrophages in the K18-hACE2 model, which lead to more than 98% depletion of the Ly6C⁻ Mo/Mφ, did not impact on disease development, control of viral replication, and induction of IFN response and *Tnfa* expression (Fig S3). Collectively, these data suggest that STING plays no essential role in the development of a COVID-19-like disease, control of SARS-CoV-2 infection, and activation of inflammation in the K18-hACE model.

In this study, we wanted to evaluate the role of STING during SARS-CoV-2 infection using an in vivo model. The STING pathway is well-known to play a protective role against many viral infections (Wu et al, 2013; Reinert et al, 2016; Flood et al, 2019). In addition, exogenous treatment with STING agonists exerts antiviral activity, but also inflammation (Skouboe et al, 2018; Uhlorn et al, 2020; Amouzegar et al, 2021). Two studies showed that the use of a STING agonist can exert potent control of SARS-CoV-2 infection both in vitro and in vivo (Humphries et al, 2021; Li et al, 2021). Upon administration of diABZI during SARS-CoV-2 infection in K18-hACE2 mice, the investigators found a decrease in viral load, induction of IFN-stimulated genes (ISGs), and recruitment of cells, including neutrophils. In our study, we did not observe any effect of STING deficiency on induction of the expression of IFN-I genes or ISGs, nor on viral load or recruitment of neutrophils. Collectively, these data suggest that although the STING-IFN pathway has the potential to control SARS-CoV-2 replication if activated in a sufficient and timely manner, it is redundant for control of infection in the K18-hACE2 model.

A recent study has have demonstrated that STING is activated in cells of COVID-19 patients and the cGAS–STING pathway was suggested that the pathway contributes to disease development (Domizio et al, 2022). Similar to what has been reported in COVID-19 patients and SARS-CoV-2-infected cells (Zhou et al, 2021; Domizio et al, 2022; Neufeldt et al, 2022), we found the STING pathway to be activated in SARS-CoV-2-infected K18 hACE2 mice. In the study by Domizio *et al*, daily administration of the STING antagonist H-151 to SARS-CoV-2-infected K18-hACE2 mice led to a reduction in inflammation and cell infiltration. The authors also reported a decrease in ISG expression but no difference in viral replication. Based on these data, it was concluded that the cGAS–STING pathway is involved in development of severe COVID-19. In contrast to the

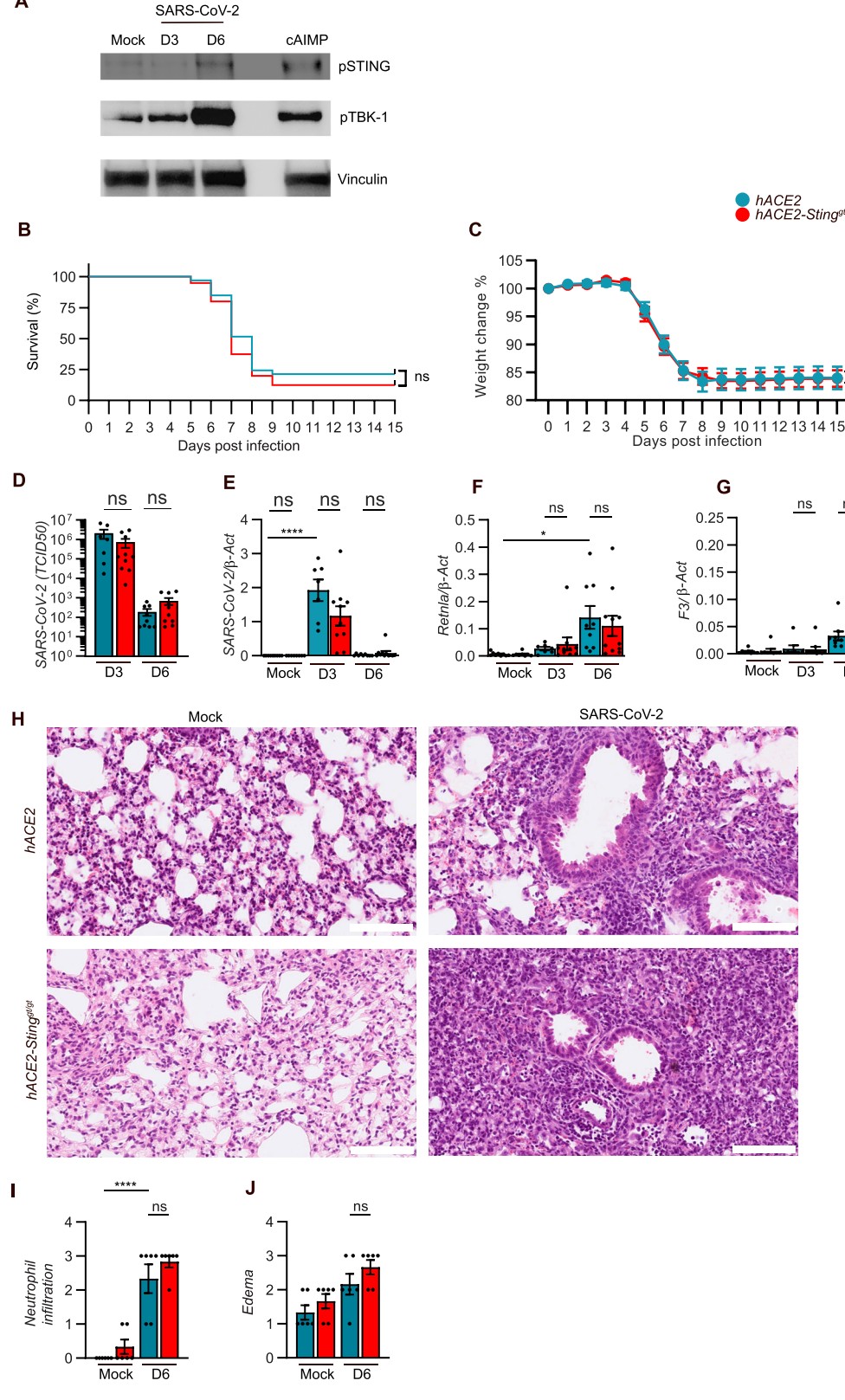

**Figure 1. Sting is not essential for increased susceptibility to SARS-CoV-2 infection.**

11–12-wk-old male and female K18-hACE2-transgenic mice and 18-hACE2 Sting[gt/gt] mice were inoculated via the intranasal route with 2.5 × 10³ p.f.u. SARS-CoV-2 (alpha strain). **(A)** Western blot analysis of pSTING, pTBK-1, and vinculin from lung tissue samples of mock, Sars-CoV-2 infected or Sting ligand, cAIMP-treated mice. **(B, C)** Survival and change weight loss in percentage of initial weight was monitored daily (combination of three independent experiments; n = 33–40 mice/group, mean ± SEM), *P*-values were calculated using Log-rank (Mantel-Cox) test (B) or two-way repeated measures ANOVA with Sidak's multiple-comparison test (C), and assigned: *P* > 0.05 (ns, not significant). **(D)** Viral titers (day 3 and 6 p.i) in the lungs of mice infected with SARS-CoV-2. Data are presented as means ± SEM, n = 7–10 mice per group, Wilcoxon rank-sum test and assigned: *P* > 0.05 (ns, not significant). **(E, F, G)** Total mRNA levels of SARS-CoV-2 or lung integrity markers (*Retnla* and F3) in harvested lungs at day 3 and 6 p.i were measured by qRT-PCR analysis. Bars represent mean ± SEM of relative gene expression levels (2^−ΔCT) and represent two independent experiments. Values were normalized to house-keeping genes *β*-actin, n = 7–11 mice in each group per experiment, *P*-values were calculated by using two-way ANOVA with Bonferroni's post hoc test and assigned: *P* > 0.05 (ns, not significant). **(H)** Representative H&E staining of lungs from mock or Sars-Cov-2–infected mice at day 6 p.i. (Scale bar = 100 *μ*m). **(I, J)** Pathological assessment of histology and quantitation of two sections of each lung from mice (one experiment, n = 6 pr groups) and *P*-values were calculated by using two-way ANOVA with Bonferroni's post hoc test and assigned: *P* > 0.05 (ns, not significant).

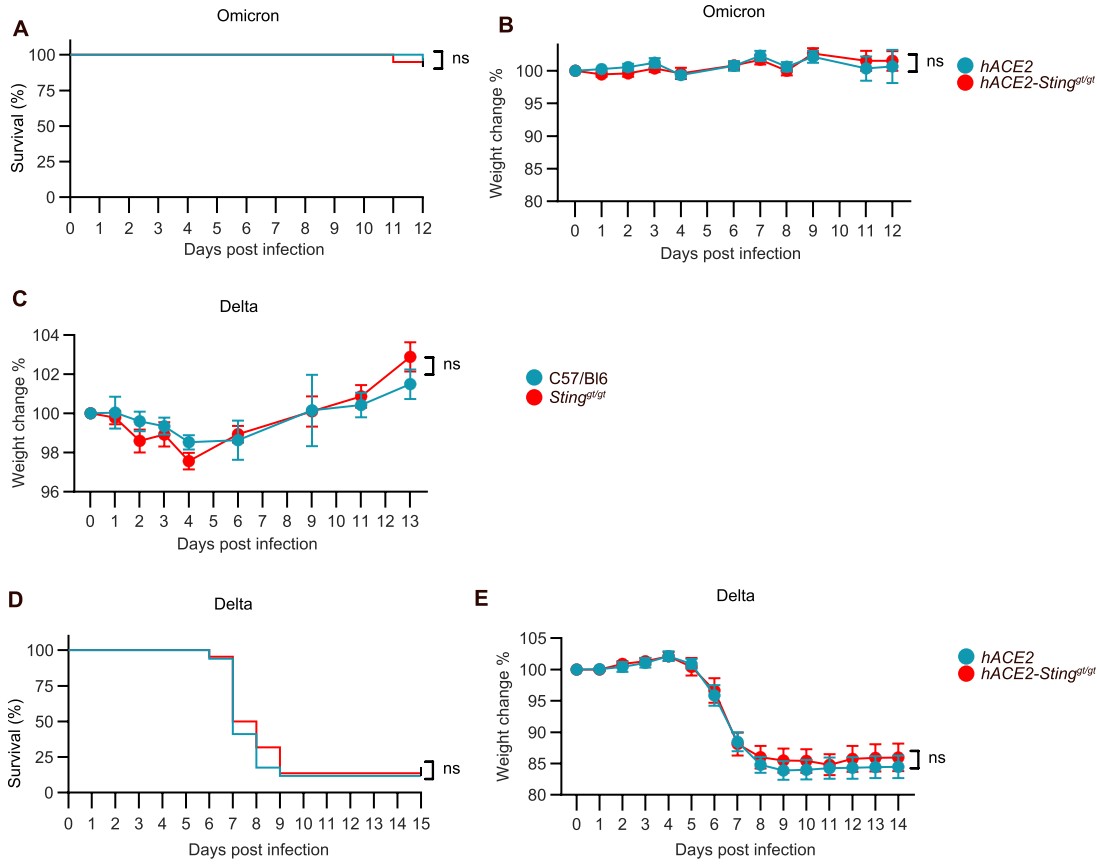

**Figure 2. Different strains were used for infections to explore differences.**
11–12-wk-old male and female K18-hACE2-transgenic mice and K18-hACE2 *Sting^gt/gt* mice were inoculated via the intranasal route with 2.5 × 10⁴ p.f.u. SARS-CoV-2 (omicron BA.1 stain). **(A, B)** Survival and change weight loss in percentage of initial weight was monitored daily (n = 12–20 mice/group, mean ± SEM) represent three independent experiments. **(C, D, E)** The mice were inoculated like in (fig A) but with 2.5 × 10⁴ p.f.u. SARS-CoV-2 (Delta strain B1.617.2 H11). Survival and change weight loss in percentage of the initial weight was monitored daily (n = 22–17 mice/group, mean ± SEM). **(A, B, C, D, E)** *P*-values were calculated using Log-rank (Mantel-Cox) test (A, C, E) or two-way repeated-measures ANOVA with Sidak's multiple-comparison test (B, C, D), and assigned: *P* > 0.05 (ns, not significant). Figures represent two independent experiments.

report from Domizio et al, our observations indicate that STING was redundant for mediating the pathological inflammatory response in COVID-19 pathogenesis. We observed that the markers of lung integrity, inflammatory cytokine expression, and leukocyte recruitment, which were profoundly altered by SARS-CoV-2 infection, were not affected by STING deficiency. Similar to the report by Domizio et al, we observed reduced levels of *Tnfrsf12a* after infection in mice with inhibited STING activity. However, we found the reduced expression of *Tnfrsf12a* to be a constitutive feature of STING-deficient mice, and the functional importance of this in COVID-19 disease remains to be explored. It is important to note that our data are based solely on the K18-hACE2 model and is therefore only a model for COVID-19, not the actual human disease. Therefore, it remains possible that STING does contribute to critical COVID-19 in humans. Interestingly, the high abundance of classical inflammatory monocytes (human, CD14⁺CD16⁺; mouse, Ly6c⁺) is observed in lungs of both critical COVID-19 patients and SARS-CoV-2–infected K18-hACE2 mice (Winkler et al, 2020; Chen et al, 2022), but we find the recruitment to be STING-independent. We did, however, observe STING-dependent the recruitment of non-classical Ly6c⁻ monocytes, which have been reported to be involved

in inflammatory disease (Misharin et al, 2014). Although this recruitment was partially STING dependent, we observed that depletion of macrophages did not affect control of SARS-CoV-2 infection or development of disease in mice. Altogether, more work is required to investigate the importance of activation of the cGAS–STING for the development of critical COVID-19 in humans. This is important for the understanding of disease pathogenesis, and to explore the therapeutic potential for cGAS or STING antagonists in COVID-19.

# Materials and Methods

### SARS-CoV2 infection murine in vivo model

K18-hACE2 mouse COVID-19 model K18-hACE C57BL/6J mice (strain: 2B6.Cg-Tg(K18-ACE2)2Prlmn/J) were obtained from The Jackson Laboratory (Stock nr: 034860). The K18-hACE2 mice were crossed with Sting^gt/gt (C57BL/6J-Sting1gt/J, Stock nr:017537) to obtain either homozygote K18-hACE2 Sting^gt/gt mice or heterozygote K18-hACE2^ki/wt

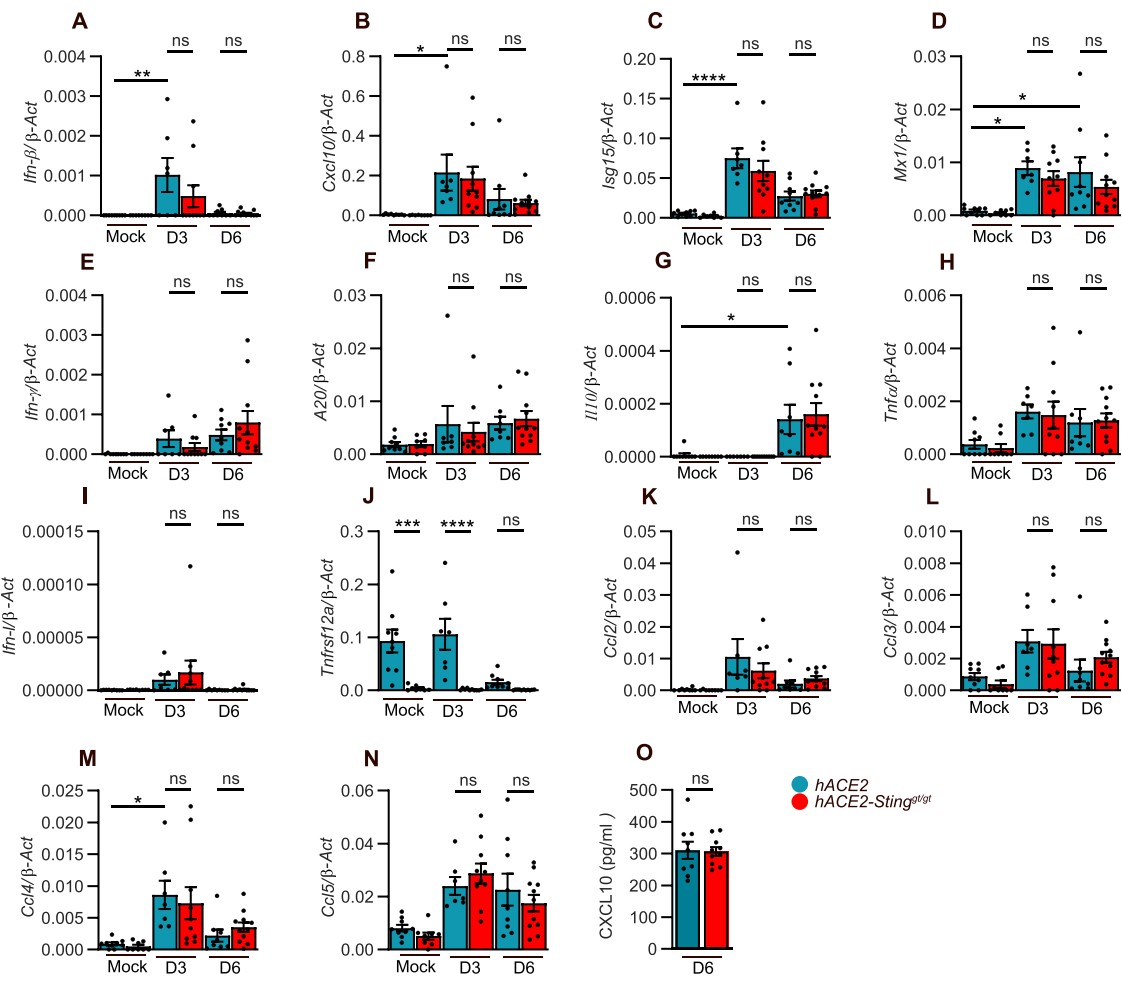

**Figure 3. Both pro- and anti-inflammatory process are independent of Sting during SARS-CoV-2 infection.**
K18-hACE2-transgenic mice and K18-hACE2 Sting[gt/gt] mice were either mock infected or infected with 2.5 × 10³ p.f.u. SARS-CoV-2 (alpha strain). **(A, B, C, D, E, F, G, H, I, J, K, L, M, N)** Lungs were harvested 3 and 6 d p.i., total mRNA were isolated, and the levels of IFN, cytokines, chemokines in lungs were measured by qRT-PCR analysis. Bars represent mean ± SEM of relative gene expression levels ($2^{-\Delta CT}$). Values were normalized to house-keeping genes $\beta$-actin, n = 7–11 mice in each group per experiment, $P$-values were calculated by using two-way ANOVA with Bonferroni's post hoc test and assigned: $P > 0.05$ (ns, not significant), *$0.01 < P < 0.05$; **$0.001 < P < 0.01$. Figures represent two independent experiments. **(O)** The lungs harvested 6 d p.i were isolated and the levels of CXCL10 were measured at protein level, the bars represent mean ± SEM.

Sting[gt/wt] mice. Age-matched male and female mice, randomized in groups, were fed standard chow diet and housed in a pathogen-free facility. Animals were anesthetized with isoflurane and administered either with 2.5 × 10³ p.f.u. SARS-CoV-2 (alpha strain) or 2.5 × 10⁴ p.f.u. SARS-CoV-2 (omicron BA.1 stain or Delta strain B1.617.2 H11) intranasally. Subsequently, the mice were placed on their backs and maintained under anesthetics for 7 min. The mice were weighed at the same time every day and euthanized in case of percentual weight loss of more than 20%, or appearance of clinical diagnostic signs of respiratory stress, including respiratory distress, which are considered humane endpoints.

For macrophage depletion, the mice were treated in vivo on day −1, 1, 3, 5 with clodronate liposomes and control liposomes (PBS) (Liposoma B.V.). The treatment was given with intranasal delivery (15 $\mu$l) and with intraperitoneal injection (200 $\mu$l).

**SARS-CoV2 TCDI50% assay**

The assay was performed as follows. 2 × 10⁴ Vero E6 TMPRSS2 cells were seeded in 90 $\mu$l DMEM (Gibco, + 2% flow cytometry standard (FCS) (Sigma-Aldrich) + 1% Pen/Strep (Gibco) + L-Glutamine (Sigma-Aldrich)) per well in flat-bottom 96-well plates. 24 h after, samples were titrated onto the cells by addition of 10 $\mu$l of a 10-fold serial dilution. One full plate was used per sample analyzed. Each dilution of supernatant was represented eight times on a plate. The cells were incubated for 72 h in a humidified $CO_2$ incubator at 37°C, 5% $CO_2$ before fixing with 5% formalin (Sigma-Aldrich) and staining with crystal violet solution (Sigma-Aldrich). Images were taken using a Leica DMi1, microscope with a Leica MC170 HD camera. TCDI50 % virus titer was calculated by Reed–Muench method.

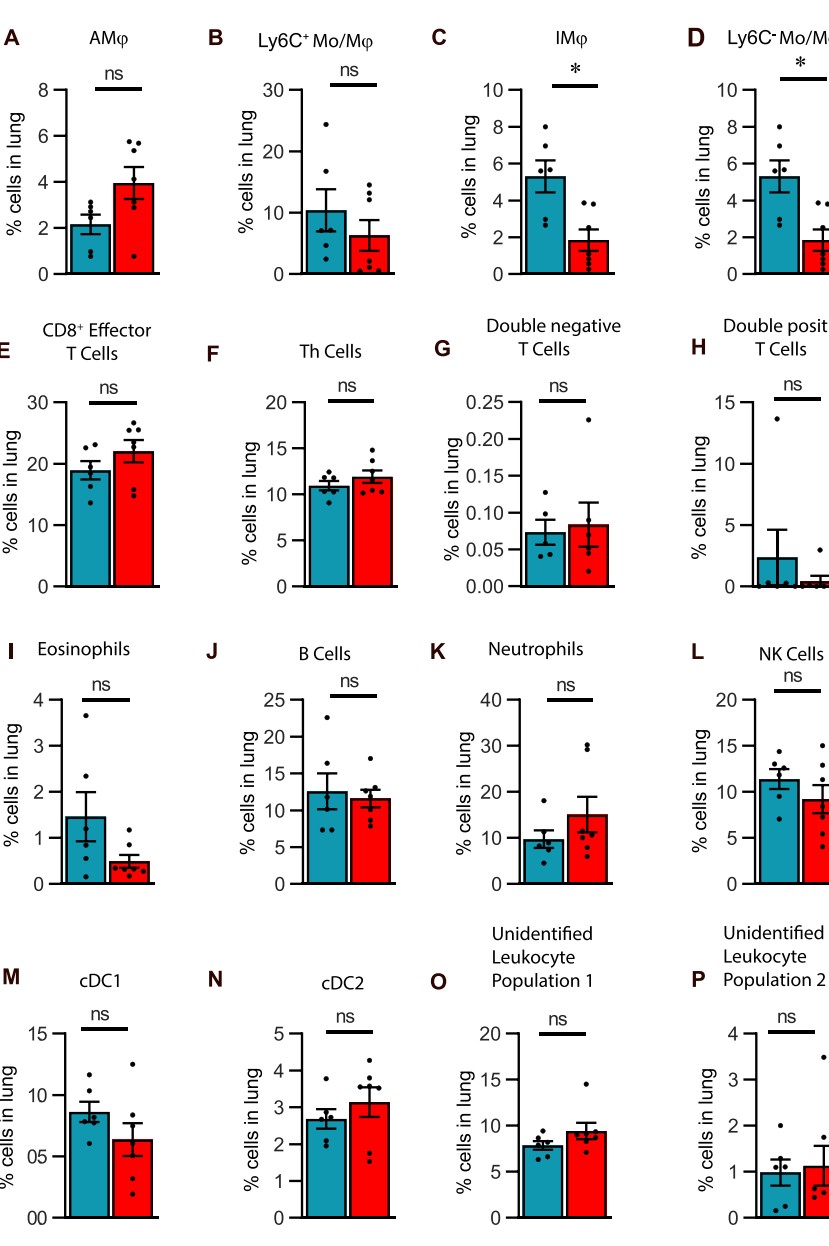

**Figure 4. Leukocyte composition and population frequencies in infected lungs of K18-hACE2-transgenic mice versus K18-hACE2 Sting<sup>gt/gt</sup> mice are not significantly changed.**

**(A, B, C, D, E, F, G, H, I, J, K, L, M, N, O, P)** K18-hACE2-transgenic mice (n = 6) and K18-hACE2 Sting$^{gt/gt}$ mice (n = 7) were infected with 2.5 × 10³ p.f.u. SARS-CoV-2 (alpha strain) and lungs were harvested at day 6 p.i Suspension mass cytometry was used to define 16 leukocyte populations based on differential expression of surface markers (UMAP clustering), and leukocyte population frequencies for each individual sample were calculated. Data represent a single experiment. Statistics: P-values were calculated by using a nonparametric Mann–Whitney test: $P > 0.05$ (ns, not significant), *$0.01 < P < 0.05$; **$0.001 < P < 0.01$. UMAP clustering and detailed phenotype of the leukocyte populations are provided as supplementary figure and supplementary table (Fig S2D). UMAP, Uniform Manifold Approximation and Projection for dimension reduction.

## RNA isolation and real-time PCR (qRT-PCR)

Lungs were homogenized with steel beads in a TissueLyser (II) (both from QIAGEN) in PBS and immediately used for RNA isolation. RNA was isolated using the High Pure RNA Isolation Kit (Roche) and an equal amount of RNA was used for standard One-Step RT–PCR (TaqMan RNA-to-Ct 1-Step Kit; Applied Biosystems). For the SARS-CoV-2 N gene, qRT-PCR primers AAATTTTGGGGAC-CAGGAAC and TGGCACCTGTGTAGGTCAAC and probe FAM-ATGTCGCGCATTGGCATGGA-BHQ were used.

For the multiplex qRT-PCR: the total RNA samples were subjected to a reverse transcription-specific target amplification step, which led to

the formation of cDNA of the mRNAs of interest. During this step, a Direct One-Step qRT–PCR kit (Thermo Fisher Scientific) and a mix of all primers of interest (TaqMan gene expression assays) were used according to the manufacturer's instructions. After testing the appropriate dilution of the pre-amplified cDNA samples, RNA quantification was achieved using the appropriate primers and TaqMan RNA-to-CT 1-Step kit (Applied Biosystems, Thermo Fisher Scientific). The Multiplex qRT-PCR was performed in the Dynamic Array Integrated Fluidic Circuits chip (Fluidigm) combining pre-amplified samples with the TaqMan primers according to the manufacturer's instructions. Taqman Gene Expression Assays were used (Applied Biosystems): F3 (Mm00438855_m1), Retnla (Mm00445109_m1), Ifn-l (Mm00663660_g1), Tnfrsf12a

(Mm07305008_m1), b-Act (Mm00607939_s1), CCL2 (Mm00441242_m1), (CCL3 (Mm00441259_g1), CCL4 (Mm00443111_m1), CCL5 (Mm01302427_m1), A20 (Mm00437121), Ifn-g (Mm01168134_m1), IL10 (Mm00439614_m1), Tnf-a (Mm00443260_g1), Mx1 (Mm00487796_m1), Isg15 (Mm01705338_s1), Cxcl10 (Mm00445235_m1), Ifn-b (Mm00439552_s1). All samples were set up in two technical replicates. The chip run was performed in a BioMark PCR Fluidigm instrument (AH Diagnostics), and the standard protocol for the Integrated Fluidic Circuits chip was followed. The data were acquired using the Fluidigm Real-Time PCR Analysis software 4.1.3 (Fluidigm). RNA levels of the gene of interest were normalized to the mouse housekeeping gene b-actin using the formula: $2^{Ct(bactin)-Ct(mRNA\ X)}$. The resulting normalized ratio is presented directly in the Figs.

### SARS-CoV-2 propagation

Clinical isolate SARS-CoV-2 B.1.1.7 SARS-CoV2 (isolate, listed in GenBank under accession number MZ314997) was provided under MTA by Professor Arvind Patel, University of Glasgow. The Omicron BA.1 strain was provided under MTA by Professor Alex Sigal, University of KwaZulu-Natal, South Africa, whereas the Delta stain (SSI-H11, B.1.617.2) wa obtained under MTA by Statens Serum Institut, København, DK. The Delta B1.617.2 H11 and Alpha variant B.1.1.7 were propagated in VeroE6 cells expressing human TMPRSS2 (VeroE6-hTMPRSS2) (kindly provided by Professor Stefan Pöhlmann, University of Göttingen), whereas the Omicron BA.1 stain was propagated in A549 cells (CCL-185; ATCC). Briefly, VeroE6-hTMPRSS2 or A549 cells were infected with a MOI of 0.05, in DMEM (Gibco) + 2% FCS (Sigma-Aldrich) + 1% Pen/Strep (Gibco) + L-Glutamine (Sigma-Aldrich) (from here, complete medium). 72 h post infection, the supernatant (containing new virus progeny) was harvested and concentrated on 100 kD Amicon ultrafiltration columns (Merck) via centrifugation at 4,000g for 30 min. Virus titer was determined using TCID50% assay and was calculated using the Reed–Muench method. To convert to the mean number of pfu/ml, the TCID50/ml was multiplied by the factor 0.7 (ATCC – Converting TCID [50] to pfu).

### Western blotting

Lungs were homogenized with steel beads in a TissueLyser (II) in 1,000 μl PBS and 100 μl were used for Western blotting. Briefly, the tissues were lysed in RIPA buffer in the presence of protease and phosphatase inhibitors then subjected to homogenization and centrifugation. The cleared supernatants were subjected to SDS–polyacrylamide gel electrophoresis and immunoblotting. The blots were blocked with 5% non-fat milk and probed with the following primary antibodies: rabbit anti-STING (1:1,000; D1V5L; Cell signaling), mouse monoclonal anti-vinculin (1:10,000; clone hVIN-1, V9131; Sigma-Aldrich), pSTING (1:1,000; D8F4W; Cell signaling), and rabbit anti-Phospho-TBK1/NAK (Ser172) (D52C2) (1:1,000; Cell signaling). Appropriate peroxidase-conjugated secondary antibodies were used for development (Jackson Immuno-Research). Secondary antibodies were as follows: anti-goat, catalogue no.705-306-147; anti-rabbit, catalogue no. 711-035-152; and anti-mouse, catalogue no.715-036-150 (all used at dilution 1:10,000).

### Histology and immunohistochemistry

Animals were anesthetized and perfused transcardially with PBS, followed by formalin at day 6 p.i. Lungs were fixed in formalin and embedded in paraffin. For routine histology, tissue sections (~4 μm each) were stained with hematoxylin and eosin. The tissue slides were assessed for the following parameters: edema, hyaline membrane formation, necrotic cellular debris, neutrophil infiltration, mononuclear inflammation, thrombosis, and hemorrhage. Criteria for assessment were largely as previously reported by Zheng et al (2021), except for thrombosis, which was assessed as either present (1) or not (0). Evaluation of mononuclear inflammation was defined as an increased amount of mononuclear inflammatory cells either diffusely present in alveolar walls or as infiltrates. Hyaline membranes and necrotic cellular debris were not observed in any mice.

### Measurement of cytokine levels

Lungs were homogenized with steel beads in a Tissue Lyser II in 1 ml PBS. 20 μl of supernatant was mixed with 80 μl 1% Triton X-100 final concentration for virus inactivation. Mouse CXCL10 level was determined by a commercially available ELISA kit (Bio-Rad) according to the manufacturer's instructions. Briefly, a 96-well microplate was coated with 50 μl per well of the diluted Capture Antibody and stored overnight at room temperature. The plate was washed three times removing any remaining liquid by inverting the plate against a clean paper towel. Each well was next filled with 150 μl of reagent diluent for blocking at room temperature for 1 h. Washing was then performed three times with wash buffer. A seven-point standard curve using twofold serial dilutions was prepared starting from 4,000 pg/ml to 62.5 pg/ml. 50 μl of the sample and standards were added per well and covered with an adhesive strip for 2 h of incubation at room temperature. The plate was then washed three times with a wash buffer. Next, 100 μl of Streptavidin-HRP was added to each well and incubated for 20 min at room temperature while avoiding direct light. 100 μl of substrate solution was added and incubated for 20 min at room temperature avoiding direct light. Before reading the plate, 50 μl of stop solution was added.

### Suspension mass cytometry

#### Preparation of specimens
Lung cell suspensions were prepared by incubating the tissue, for 1.5 h at 37°C, in tissue digestion cocktail. Tissue digestion cocktail (per sample): 900 μL RPMI (Gibco) (with Ca2+ and Mg2+); 1 mg/ml Collagenase I (Roche); 10 U/ml Pulmozyme-Dornase alpha (Roche). 20 μl 0,5 M EDTA was added to stop the enzymatic reaction. The cells were resuspended in 1,000 μl TheraPEAK ACK Lysing Buffer (10-548E; Lonza) to lyse red blood cells for 5 min. The cells were filtered through a the pre-wet 40 μM nylon mesh with PBS, centrifuged for 8 min at 300g 4°C, and resuspended in CSB. The cells were counted (TC20 automated cell counter; Bio-Rad). Up to 3mio cells/sample were washed with PBS, incubated with Cisplatin for dead cell exclusion (Cell-ID Cisplatin; SBT), (0 25 μM final concentration, 5 min incubation at RT); quenched with CSB, and subjected to Palladium-based barcoding following the

manufacturer's protocol (Cell ID 20-Plex Pd –BarcodingKit; SBT). Barcoded cells were washed, pooled, counted, and resuspended in CSB at a concentration of 3 mio cells/100 $\mu$l. Before immune staining, cells were incubated for 10 min with purified rat anti-mouse CD16/CD32 (Mouse BD Fc Block). Cell surface staining was conducted following the manufacturer's protocol (Maxpar Cell Surface Staining with Fresh Fix protocol; SBT) using a 26 surface marker antibody panel (antigen/clone/metal): B220/RA36B2/176Yb, CD11b/M1/70/148Nd, CD11c/N418/142Nd, CD14/Sa 14-2/156Gd, CD19/6D5/149Sm, CD206/C068C2/169Tm, CD24/M1/69/150Nd, CD36/HM36/147Sm, CD38/90/171Yb, CD3e/1452C11/152Sm, CD4/RM4-5/145Nd, CD43/S11/146Nd, CD45/30F11/89Y, CD64/X54-5/7.1)/151Eu, CD86/GL1/172Yb, CD8a/536.7/168Er, CX3CR1/SA011F11/164Dy, EpCAM(CD326)/G8.8/166Er, F4_80/BM8/159Tb, Ly6C/HK1.4/162Dy, Ly6G/1A8/141Pr, MHCII/M5/114.15.2/174Yb, NK1.1(CD161)/PK136/165Ho, Siglec-F/E50-2440/153Eu, TCRb/H57-597/143Nd,TER-119/TER119/154Sm. Metal-labeled Abs were obtained from SBT, except Siglec-F (BD), that was custom labeled using the MaxPar X8 labeling kit (SBT) according to manufacturer's instructions (SBT) After staining and washing, the cells were fixed with 1.6% formaldehyde for 15 min and DNA-stained over night at 4°C with 250 nM of Cell- IDTM Intercalator-IR (SBT) in FixPerm Buffer (SBT). On the next day, cells were prepared for long-term storage by aliquoting 2–3mio cells into fresh tubes, followed by centrifugation, removal of the supernatant (about 100 $\mu$l residual volume), resuspension, and storage at –80°C.

### Sample acquisition

For sample acquisition, stored cell aliquots were thawed, washed 1x with CSB, 2x with CAS, counted, and kept as pellet at 4°C. Immediately before sample acquisition, cells were resuspended in EQ Four Element Calibration Beads (SBT) diluted in CAS at a concentration of 1 mio/ml. Sample acquisition was performed on a CyTOF Helios instrument (SBT) at the Aarhus university mass cytometry unit (MCU).

### Data analysis

CyTOF datasets were exported as FCS files, randomized, normalized, concatemerized, and de-barcoded according to the manufacturer's instructions (CyTOF software Version 7; SBT). FCSExpress (Version 7; DeNovoSoftware) was used for all subsequent data analysis. Single-data files were cleaned using Gaussion distribution parameters and gated on DNA-positive (Ir positive), viable (Cisplatin neg), Ter119-negative, and CD45-positive events.

From each sample, 45.000 CD45$^+$ cells were merged and used for UMAP dimensionality reduction (number of neighbours:15; min low dim distance: 0.1; number of iterations: 500). UMAP populations were subsequently manually gated. Marker expression level and frequencies of UMAP populations from single datafiles were exported and analyzed using Microsoft ExCel.

SBT: Standard BioTools; CSB: Cell Staining Buffer (SBT); CAS: Cell Acquisition Buffer (SBT). UMAP = Uniform Manifold Approximation and Projection for Dimension Reduction.

### Ethics

The Danish Animal Experiments Inspectorate has approved the experimental animal procedures and they were carried out in accordance with the Danish Animal Welfare Act for the Care and Use of Animals for Scientific Purposes (License ID 2019-15-0201-00090 and 2020-15-0201-00726). All procedures followed the recommendations of the Animal Facilities at the Universities of Copenhagen and Aarhus.

## Data Availability

The data that support the findings of this study are available from the corresponding author upon request.

## Supplementary Information

## Acknowledgements

The technical assistance of Ea Stoltze Andersen, Kirsten Stadel Petersen, and Marie Beck Iversen was greatly appreciated. The work was funded by The Independent Research Fund Denmark (0214-00001B), the European Research Council (ERC-AdG ENVISION; 786602), and the Novo Nordisk Foundation (NNF18OC0030274).

### Author Contributions

G Marino: data curation, formal analysis, validation, investigation, methodology, and writing—original draft, review, and editing.
B Zhang: data curation, validation, investigation, visualization, methodology, and writing—original draft.
A Schmitz: data curation, software, formal analysis, validation, investigation, visualization, methodology, and writing—review and editing.
HVF Schwensen: data curation, formal analysis, validation, visualization, methodology, and writing—review and editing.
LS Reinert: conceptualization, data curation, formal analysis, supervision, validation, investigation, visualization, methodology, project administration, and writing—original draft, review, and editing.
SR Paludan: conceptualization, formal analysis, supervision, funding acquisition, validation, investigation, visualization, project administration, and writing—original draft, review, and editing.

### Conflict of Interest Statement

The authors declare that they have no conflict of interest.

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
