## [Reviewer comments · Life Science Alliance]

Life Science Alliance

STING is redundant for host defense and pathology of COVID-19-like disease in mice

Giorgia Marino, Bao-cun Zhang, Alexander Schmitz, Hanna Schwensen, Line Reinert, and Søren Paludan

DOI: <https://doi.org/10.26508/lsa.202301997>

Corresponding author(s): Søren Paludan, Aarhus University; Line Reinert, Aarhus University; and Søren Paludan, Aarhus University

Review Timeline:

Submission Date:	2023-02-17
Editorial Decision:	2023-02-22
Revision Received:	2023-05-20
Editorial Decision:	2023-05-22
Revision Received:	2023-05-25
Accepted:	2023-05-25

Transaction Report:

Please note that the manuscript was previously reviewed at another journal and the reports were taken into account in the decision-making process at *Life Science Alliance*.

Referee #1 Review

Report for Author:

The role of DNA sensing in RNA viral infection is an interesting and poorly understood process. Here the authors have infected K18 hAce2 mice with SARS-CoV-2 and measured a series of markers and features of disease comparing wild type and STING KO mice. Despite measuring a series of markers of infection driven immune activation and inflammation, the authors only found a few that differed between the wild type K18 hAce2 and STING KO mice. They conclude that STING does not have a role for SARS-CoV-2 induced disease in this mouse model.

Although the authors conclude no role for cGAS they do see differences in specific measurements, for example, Fig3J (which isn't discussed) as well as some specific macrophage markers in Fig 4. The authors conclude these macrophage measurements do not contribute to disease because they observe similar levels of disease in the STING KO mice. Additionally they show that depleting macrophages with clodronate also does not impact disease, as far as they measure this by cytokines PCR and weight loss.

I'm afraid I'm not persuaded by the author's conclusions because I do not understand exactly how the authors are defining disease. Many measurements are made, and STING KO impacts some but not others. I would argue that if any immune related markers are different when STING is knocked out, then the response to CoV-2 infection, ie disease, is changed. I assume the authors are referring to overt disease but its very unclear how overt disease is judged. In the methods it says "Mice were weighed every day at the same time of the day and in case of percentual weight loss of more than 20 percent or clinical signs of more than moderate suffering were considered as humane endpoint." However, I don't see evidence that any mice did lose more than 20% of their weight so the Kaplan meyer plots must have been based on judgement of symptoms in a way that is not described.

Basically, the authors haven't seen gross changes in overt disease but they have seen changes in macrophage activation when they KO STING. I don't think its safe to conclude, given the measurements made, that there is no impact on disease. I think this is particularly true when there are 2 other papers, which are referenced, that propose that cGAS detection of e.g. mitochondrial DNA does have a role in COVID19 disease. Further, Domizio et al link this to "macrophages adjacent to areas of endothelial cell damage" (from abstract). One could argue that this conclusion is related to the differences measured herein. If the authors feel their data clearly demonstrates these other studies to be wrong, they should explain why. But they don't really do that and I think that's because their study is simply not performed in enough depth to make that case. I don't think its reasonable to suggest there's some controversy in this area by performing a less in depth study that actually makes related findings.

1. I propose a more honest discussion of the data and a more in depth discussion of how the data presented differ from these 2 previous studies and how we should interpret that and a title that more accurately reflects the data.

2. The authors should also discuss Fig 3J.

3. Page 6 para 2. I don't think we say data not shown any more. Please show or remove.

Referee #2 Review

Report for Author:

In their presented manuscript, the authors investigate the involvement of the cGAS/STING pathway onto COVID-19-like disease in the K18-hACE2 mouse model. The authors observe no significant difference in classical disease parameters, such as weight loss and mortality, neither do they find differences in lung cytokine levels or immune cell composition (apart from non-classical monocytes and interstitial macrophages). They conclude that STING, albeit activated upon SARS-CoV-2 infection in the lung, does not impact on the development and progression of the disease. The important specification to this "bottom line" is rightfully brought up by the authors themselves: it only holds for this one very particular model, but might not reflect pathogenesis in more relevant animal models or even the human situation.

The manuscript is very well written and nicely presented. The experiments appear to be performed to highest standards. Still, although I very much like the concise and straight-forward manuscript, I do not see a high degree scientific impact and significance, as it exclusively focusses on this one (rather artificial) mouse model.

My recommendation would be to consider a more specialized outlet for the manuscript as is, or invest significantly into establishing the patho-mechanism of SARS-CoV-2 in the K18 model and/or move to a more physiological model system and only perform confirmatory experiments in K18. My main concern in this regard is the question whether inflammatory dysregulation and in particular the IFN response indeed are the driving forces behind disease development in K18 mice. It is well established that morbidity and lethality of SARS-CoV-2 infection in K18 is dominated by the unnatural neurotropism in this model (due to high hACE2 expression in the brain). Fumagalli et al (Science Immunology 2021, 10.1126/sciimmunol.abl9929) establish infection via aerosols (opposed to standard intranasal infection) to overcome this severe limitation. Another possibility would be to employ mouse-adapted virus in wild type animals (e.g. as in Chong et al. 10.1016/j.celrep.2022.110799, or Beer et al. 10.1084/jem.20220621). In the present manuscript, the authors even perform one experiment in wild type C57/Bl6 animals with the delta variant. Here, however, they observe a very unusual mild pathology with hardly any weight loss- it is, hence, impossible to compare STING-KO vs wild type here; likely optimizing the experimental system (higher virus dose?) might be worthwhile. A last option I could envisage would be to investigate K18 hACE2 IFNAR/IFNLR-KO animals to prove that indeed IFN induction does play a role for development of disease in this model, as has been well established in humans.

As it stands, I have my doubts whether the results of the (nicely and solidly performed!) experiments in this manuscript can easily be transferred to the human situation (or even other animal models). This substantially limits the scope of the study and, accordingly, the interest of a broader readership.

Referee #3 Review

Report for Author:

The study of Marino et al. investigates the role of STING pathway following SARS-CoV-2 infection. To address this question, authors used a genetic approach, using K18-hACE2 (a classic mouse model for Sars-CoV2 infection) invalidated for STING. Their results show that, in the absence of STING, K18-hACE2 are as susceptible to SARS-CoV2 infection (in terms of survival, viral load, cytokine production and lung pathology), as compared to STING-sufficient controls. Authors thus conclude that the role of STING in vivo is redundant, in response to SARS-Co-V2 infection.

Interestingly, previous studies have shown some discrepancies regarding the role of cGAS-STING pathway in SARS-CoV2 associated physiopathology in vivo (either antiviral or promoting lung pathology and cytokine storm, in response to SARS-CoV2 infection). As a result, in those previous studies, both pharmacological inhibition and activation of STING conferred SARS-CoV2 infected mice a better outcome.

Altogether, the study of Marino et al. appears very informative to the field, since STING has been considered as a therapeutic target in Covid patients. Moreover, authors have clearly made some substantial efforts in terms of data reproducibility (e.g. number of animals per group and independent experiments), which is worth pointing out considering the challenges of in vivo work. However, most experimental settings (where none of the K18-hACE2-WT mice survive infection) mainly question the role of STING in the overt-inflammatory state associated with SARS-CoV2 infection, with the hypothesis that, in the absence of STING, there would be less inflammation and mice survival rate would increase. The study would largely benefit from investigating the putative beneficial role of STING as well (namely to use an experimental model allowing to evaluate a decrease in mice survival, in the absence of STING). Finally, and although beyond the scope of this study, authors should consider STING inducible KO mice in future studies, to investigate the role of STING (potentially different) at different time post SARS-Co-V2 infection.

Major comments:

1: Fig 1 and corresponding text:

Papers from Domizio et al. and Li et al. (references cited by the authors) appears opposite in their conclusions regarding the role of STING in vivo, following SARS-Co-V2 infection. Domizio et al. shows that pharmacological inhibition of STING reduces lung pathology, inflammation and weight loss, altogether leading to a slight increase in survival rate. By contrast, Li et al. shows that pharmacological activation of STING leads to decreased viral load and weight loss and overall survival of mice. In the experimental models from the present study, with SARS-Co-V2 being lethal in K18-hACE2 WT

animals, authors can only appreciate a putative beneficial impact of STING KO, on mice survival. However, one cannot exclude that genetic ablation of STING could have a negative impact on animals, in response to SARS-Co-V2 infection. While one would appreciate the difficulty to have a mouse model for a sublethal dose of SARS-Co-V2, it is interesting to note that, in FigEV1 F and G, the use of K18-hACE2 heterozygous mice only confers about 50% of mortality, in STING sufficient animals. However, in those experiments, authors used mice heterozygous for STING, which makes it difficult to assess rigorously the role of STING in this system.

Altogether, the authors are invited to compare survival and weight loss of K18-hACE2(HET) and K18-hACE2(HET)-STING KO (Ho), following SARS-Co-V2 infection. Additional sets of experiments to look at lung histopathology and cytokine production could also be useful in that setting, to fully exclude a redundant role of STING in vivo. Such a system would also allow to perform some analysis at later time-points than day 6 post infection, to study additional features such as T-cells exhaustion, mentioned in the introduction.

2- Figure 1- F&G and corresponding text:

Authors are invited to show some analysis of lung histopathology (e.g. H&E staining) to complement the qPCR analysis.

3- Figure 2 and corresponding text:

Although infection with Omicron variants is not lethal in mice, some studies could show some degree of clinical features and symptoms (Halfmann et al. (<https://doi.org/10.1038/s41586-022-04441-6>; Tarrés-Freixas et al. <https://doi.org/10.3389/fmicb.2022.840757>). In line with comment 1, it would be very interesting to assess viral loads and lung pathology following Omicron and Delta infections, in both "WT" (C57Bl/6 or K18-hACE2) and "STING-KO" mice.

4- Figure 3 and corresponding text:

To complete this analysis, the authors are invited to confirm qPCR results at the protein level (e.g. Elisa in bronchoalveolar fluids from infected mice).

5- Figure 4: If additional experiments are performed in a "non-lethal" context for WT animals (cf comment 1), it could be useful to perform an intravascular staining before animal killing, for the study of immune cells in the lung (Anderson et al, 10.1038/nprot.2014.005). Since the lung is highly vascularized, it could help identifying cells that are truly within lung tissue.

Minor points:

6- Introduction (First paragraph): While mentioning SARS-Co-V2 cell entry receptors, it would be nice to mention other cell entry mechanism, such as TMPRSS2, which appears important, especially in mice (e.g. <https://doi.org/10.1038/s41467-022-33911-8>)

7- Figure 4 and EV2, and corresponding text:

7.1 (Figure 4): Absolute numbers could be shown (if available), rather than % of cells populations. Absolute numbers are generally more informative: for example, % could be the same but with a lower number of total cells to begin with.

7.2 (Figure 4 - G & H): It is quite surprising to see some DN or DP T-cells (representing stages of T cells development) in the lung of infected animals. In graphs G and H, it seems that the error bars are strongly impacted by 1 dot (namely 1 mouse) being DN or DP, while all the other animals are clearly negative for those sub-populations. Authors are invited to double-check the raw data of the corresponding mice, for any issue in the samples.

7.3 (Figure 4 O & P and Figure EV2-D): Authors should specify for which markers "unidentified leukocytes" are negative for.

8- Figures' legends: Author should mention the number of independent experiments performed for each figure, in figure legend.

9- Material and Methods, section "SARS-Co-V2 infection murine in vivo model":

Authors should specify which viral dose has been used for each Sars-CoV2 variant, in material and methods.

February 22, 2023

Re: Life Science Alliance manuscript #LSA-2023-01997-T

Prof. Søren R Paludan
Aarhus University
Biomedicine
Wilhelm Meyers Alle
Aarhus 8000
Denmark

Dear Dr. Paludan,

Thank you for submitting your manuscript entitled "STING is redundant for host defense and pathology of COVID-19-like disease in mice" to Life Science Alliance. We invite you to submit a revised manuscript addressing the following Reviewer comments:

- Address Reviewer 1's comments.
- Address Reviewer 3's major comments #2 & 4 and minor points.

Thank you for this interesting contribution to Life Science Alliance. We are looking forward to receiving your revised manuscript.

Sincerely,

- A letter addressing the reviewers' comments point by point.
- An editable version of the final text (.DOC or .DOCX) is needed for copyediting (no PDFs).
- High-resolution figure, supplementary figure and video files uploaded as individual files: See our detailed guidelines for preparing your production-ready images, <https://www.life-science-alliance.org/authors>
- Summary blurb (enter in submission system): A short text summarizing in a single sentence the study (max. 200 characters including spaces). This text is used in conjunction with the titles of papers, hence should be informative and complementary to the title and running title. It should describe the context and significance of the findings for a general readership; it should be written in the present tense and refer to the work in the third person. Author names should not be mentioned.
- By submitting a revision, you attest that you are aware of our payment policies found here: <https://www.life-science-alliance.org/copyright-license-fee>

B. MANUSCRIPT ORGANIZATION AND FORMATTING:

We thank both reviewers for their positive comments and for the suggestions that have helped us to improve the clarity and completeness of our manuscript. Our point-by-point reply follows.

Referee #1:

1. I propose a more honest discussion of the data and a more in depth discussion of how the data presented differ from these 2 previous studies and how we should interpret that and a title that more accurately reflects the data.

Response: As pointed out by the reviewer, our data show that STING does not play a role in SARS-CoV2 infection in mice. In contrast to our study with full KO of STING, Domizio et al. did transient chemical inhibition of STING activity by injecting mice daily with a compound dissolved in either dimethyl sulfoxide (DMSO) or PBS 5% or Tween-80, where they showed less inflammation and less weight loss in mice which did not have activation of STING. Comparable to our study they did not observed any difference in viral load in the lungs after day 3 and 6 p.i. In our study we additionally conclude that STING is redundant for disease development. The reviewer is right that many measurements are made to describe the disease but were not explained in sufficient details in the original manuscript. The overall disease course is now fully defined in the M&M. The only two differences we observed between WT and KO mice are; (1) recruitment of non-classical Ly6c- monocytes was partially STING-dependent and the levels of ontologically linked interstitial macrophages was lower in STING-deficient mice. However, we observed no effect of depletion of macrophages on control of SARS-CoV-2 infection or development of disease in mice. (2) The expression of *Tnfrsf12a* was constitutively lower in mice lacking STING. At this point we cannot explain why this TNF receptor superfamily member is constitutively downregulated in the KO mice. We have included a note on this in the discussion of the revised manuscript.

With only the above two parameters being affected by STING-deficiency among the very large number of parameters measured (including both clinical and preclinical parameters), it is fair to conclude that STING is redundant for host defense and pathology during SARS-CoV2 infection in mice.

Our study does not demonstrate - neither claims - the other studies published to be wrong. It is very difficult to compare our data with studies with KO mice from Domizio et al and Li et al. where they investigate a role for STING upon administration of compounds to either over-activate or down-regulate STING. We cannot exclude a potential role for STING in controlling SARS-CoV-2 disease if activated in a sufficient and timely manner. We are very specific on this point in the manuscript. What can be concluded from our study, is that the role of STING in the pathogenesis of critical COVID-19 remains unresolved, and requires further investigation.

2. The authors should also discuss Fig 3J.

Response: We thank the reviewer for noticing this and this is now included in the discussion.

3. Page 6 para 2. I don't think we say data not shown any more. Please show or remove [Red: *delta variant infection*].

Response: We now show the data from the experiments with SARS-CoV-2 delta variant infection in the K18-hACE2 mouse model, which are included in the revised manuscript as Fig 2C-D. Consistent with the data from with other virus variants, we also observed no effect of STING-deficiency on disease development and survival.

Referee #3:

Major comments:

2- Figure 1- F&G and corresponding text:

Authors are invited to show some analysis of lung histopathology (e.g. H&E staining) to complement the qPCR analysis.

Response: We have now included a pathological examination of our mouse model. The STING independent phenotype findings are confirmed by H&E stainings of the lungs where we observed significantly elevated neutrophil infiltration, hemorrhage and thrombosis in the infected lungs compared to mock infected lungs (Fig. 1H-J and EV1 H). We did not observe any difference in mononuclear infiltrates, hemorrhage, thrombosis, edema or neutrophil infiltration in K18-hACE and STING-deficient K18 h-ACE2 mice.

4- Figure 3 and corresponding text:

To complete this analysis, the authors are invited to confirm qPCR results at the protein level (e.g. Elisa in bronchoalveolar fluids from infected mice).

Response: As suggested by the reviewer we have now performed ELISA analysis of the lung homogenate and measured one of the highly expressed ISGs, CXCL10 at the protein level. We observed no significant difference between samples from SARS-CoV-2-infected K18-hACE and STING-deficient K18 h-ACE2 mice (Fig. 3O).

Minor points:

6- Introduction (First paragraph): While mentioning SARS-Co-V2 cell entry receptors, it would be nice to mention other cell entry mechanism, such as TMRSS2, which appears important, especially in mice (e.g. <https://doi.org/10.1038/s41467-022-33911-8>)

Response: We thank the reviewer for refining our manuscript with this detail. This is now included.

7- Figure 4 and EV2, and corresponding text:

7.1 (Figure 4): Absolute numbers could be shown (if available), rather than % of cells populations. Absolute numbers are generally more informative: for example, % could be the same but with a lower number of total cells to begin with.

Response: We have now included this information I S2

7.2 (Figure 4 - G & H): It is quite surprising to see some DN or DP T-cells (representing stages of T cells development) in the lung of infected animals. In graphs G and H, it seems that the error bars are strongly impacted by 1 dot (namely 1 mouse) being DN or DP, while all the other animals are clearly negative for those sub-opulations. Authors are invited to double-check the raw data of the corresponding mice, for any issue in the samples.

Response: Thank you for noticing this. We see that the same 2 mice (one in KO and one in the control group) are giving the high values in both DN and DP, and it turns out that they are outliers. These are now removed in the revised figure.

7.3 (Figure 4 O & P and Figure EV2-D): Authors should specify for which markers "unidentified leukocytes" are negative for.

Response: The unidentified leukocytes are CD45+ cells, but did not have any of the combinations of the markers used in (fig.4 A-N).

8- Figures' legends: Author should mention the number of independent experiments performed for each figure, in figure legend.

Response: This information is now included.

9- Material and Methods, section "SARS-Co-V2 infection murine in vivo model":

Authors should specify which viral dose has been used for each Sars-CoV2 variant, in material and methods.

Response: This information was originally included in the Figure legend, but is now also included in the Material and Methods. Animals were anesthetized with isoflurane administered intranasally with either 2500 p.f.u. SARS-CoV-2 (alpha strain) or 25000 p.f.u. SARS-CoV-2 (omicron BA.1 stain or Delta strain B1.617.2 H11)

May 22, 2023

RE: Life Science Alliance Manuscript #LSA-2023-01997-TR

Prof. Søren R Paludan
Aarhus University
Biomedicine
Wilhelm Meyers Alle
Aarhus 8000
Denmark

Dear Dr. Paludan,

Thank you for submitting your revised manuscript entitled "STING is redundant for host defense and pathology of COVID-19-like disease in mice". We would be happy to publish your paper in Life Science Alliance pending final revisions necessary to meet our formatting guidelines.

- please upload your manuscript text as a doc file
- please upload both your main and supplementary figures as separate single files
- please add ORCID ID for secondary corresponding author-they should have received instructions on how to do so
- please add the Twitter handle of your host institute/organization as well as your own or/and one of the authors in our system
- please add the author contributions to the main manuscript text

Figure Check:

- please add a scale bar to Figure 1H

A. FINAL FILES:

B. MANUSCRIPT ORGANIZATION AND FORMATTING:

Sincerely,

May 25, 2023

RE: Life Science Alliance Manuscript #LSA-2023-01997-TRR

Prof. Søren R Paludan
Aarhus University
Biomedicine
Wilhelm Meyers Alle
Aarhus 8000
Denmark

Dear Dr. Paludan,

Thank you for submitting your Research Article entitled "STING is redundant for host defense and pathology of COVID-19-like disease in mice". It is a pleasure to let you know that your manuscript is now accepted for publication in Life Science Alliance. Congratulations on this interesting work.

DISTRIBUTION OF MATERIALS:

Again, congratulations on a very nice paper. I hope you found the review process to be constructive and are pleased with how the manuscript was handled editorially. We look forward to future exciting submissions from your lab.

Sincerely,
